# Evaluating a Novel Approach to Detect the Vertical Structure of Insect Damage in Trees Using Multispectral and Three-Dimensional Data from Drone Imagery in the Northern Rocky Mountains, USA

Abhinav Shrestha [1,*], Jeffrey A. Hicke [1], Arjan J. H. Meddens [2], Jason W. Karl [3] and Amanda T. Stahl [2]

1. Department of Earth and Spatial Sciences, University of Idaho, Moscow, ID 83844, USA; jhicke@uidaho.edu
2. School of the Environment, Washington State University, Pullman, WA 99164, USA; arjan.meddens@wsu.edu (A.J.H.M.); atstahl@wsu.edu (A.T.S.)
3. Department of Forest, Rangeland, and Fire Sciences, University of Idaho, Moscow, ID 83844, USA; jkarl@uidaho.edu
* Correspondence: abhinav.shrestha96@gmail.com

**Abstract:** Remote sensing is a well-established tool for detecting forest disturbances. The increased availability of uncrewed aerial systems (drones) and advances in computer algorithms have prompted numerous studies of forest insects using drones. To date, most studies have used height information from three-dimensional (3D) point clouds to segment individual trees and two-dimensional multispectral images to identify tree damage. Here, we describe a novel approach to classifying the multispectral reflectances assigned to the 3D point cloud into damaged and healthy classes, retaining the height information for the assessment of the vertical distribution of damage within a tree. Drone images were acquired in a 27-ha study area in the Northern Rocky Mountains that experienced recent damage from insects and then processed to produce a point cloud. Using the multispectral data assigned to the points on the point cloud (based on depth maps from individual multispectral images), a random forest (RF) classification model was developed, which had an overall accuracy (OA) of 98.6%, and when applied across the study area, it classified 77.0% of the points with probabilities greater than 75.0%. Based on the classified points and segmented trees, we developed and evaluated algorithms to separate healthy from damaged trees. For damaged trees, we identified the damage severity of each tree based on the percentages of red and gray points and identified top-kill based on the length of continuous damage from the treetop. Healthy and damaged trees were separated with a high accuracy (OA: 93.5%). The remaining damaged trees were separated into different damage severities with moderate accuracy (OA: 70.1%), consistent with the accuracies reported in similar studies. A subsequent algorithm identified top-kill on damaged trees with a high accuracy (OA: 91.8%). The damage severity algorithm classified most trees in the study area as healthy (78.3%), and most of the damaged trees in the study area exhibited some amount of top-kill (78.9%). Aggregating tree-level damage metrics to 30 m grid cells revealed several hot spots of damage and severe top-kill across the study area, illustrating the potential of this methodology to integrate with data products from space-based remote sensing platforms such as Landsat. Our results demonstrate the utility of drone-collected data for monitoring the vertical structure of tree damage from forest insects and diseases.

**Keywords:** drone remote sensing; structure-from-motion; point cloud; forest health monitoring; top-kill

## 1. Introduction

Forests are integral components of the biosphere that provide ecosystem services such as habitats, nutrient and water cycling, absorption of air pollutants, and carbon

sequestration [1,2]. From economic and societal standpoints, the maintenance of forests ensures the sustainable supply of harvested goods (e.g., timber, fuel, and bioproducts), and many communities value forests as spaces for cultural traditions and recreation [3,4]. Forests provide a diverse range of benefits to humans and the ecosystem; hence, monitoring the tree health status is important for the sustainable management of forests.

Insect disturbances account for a large fraction of tree mortality in many forest ecosystems [5–7], including forests in the western United States [8]. Insect outbreaks play a vital role in the functioning of forest ecosystems. However, anthropogenic climate change has facilitated more extensive outbreaks via faster insect development, greater winter survival, and increased host stress, with continued impacts expected in the future [9–14].

Remote sensing offers a wide range of data which are available at various spatial and temporal scales and resolutions [15], allowing researchers and management personnel to tailor their data collection requirements to the scope of their program objectives. In the case of the detection and monitoring of forest disturbances, remote sensing can provide a cost-effective alternative to supplement traditional labor-intensive field data collection over large spatial extents due to its lower cost of data acquisition per unit area [12,16]. As such, remote sensing can be useful for the monitoring of insect outbreaks over large areas at fine resolutions, as tree damage and mortality are often sparse [17], and thus, difficult to capture through field surveys alone.

Forest insect outbreaks can be detected through the remote sensing of characteristic traits associated with tree damage and mortality caused by insects (e.g., spectral or thermal signatures) [15,18]. For example, the USDA Forest Service (USFS) identifies visible traits in aerial detection surveys (ADSs) for assigning the following damage classes to forest insect outbreaks: mortality, discoloration, dieback, branch flagging, branch breakage, mainstem broken or uprooted, defoliation, and top-kill [19]. Classes such as these can be used to attribute damage to specific biotic damage agents and stages of infestation. For example, top-kill is defined as the death of branches in the upper parts of the crown [19,20]. Top-kill occurs in the early infestation stages of the Douglas-fir tussock moth (DFTM) (*Orgyia pseudotsugata*) [21], western spruce budworm (WSBW) (*Choristoneura freemani*) [19,22], and fir engraver (*Scolytus ventralis*) [23] and is an indicator of the severity of defoliation [24,25].

In recent years, the availability of relatively inexpensive uncrewed aerial systems (UASs, herein referred to as "drones") has led to an increase in forest ecology studies using drone-based sensors and photogrammetry techniques [16,26]. The detection and monitoring of insect infestations is a prominent area of focus for drone-based forest remote sensing research, as predicted by Hall et al. [12] and Senf et al. [17]. In their review of studies monitoring forest health using drone-based remote sensing, Ecke et al. [27] stated that most focused on the remote sensing of insect outbreaks.

The products from drone-based remote sensing and photogrammetry can be broadly categorized into two groups: point cloud (three-dimensional (3D)) data and imagery (two-dimensional (2D)) data. Point cloud data provide the structural representations of objects such as trees [28] and are useful in providing insights into forest ecosystems and disturbances [27,29,30]. Drone-based remote sensing with active sensors such as light detection and ranging (lidar) provide point clouds [30], yet they can be relatively expensive compared to the more common passive sensors such as multispectral and red–green–blue (RGB) sensors [29]. The data from these passive sensors are stored as imagery containing 2D attributes [28]. In addition to their use as images, advances in computer vision and image processing algorithms such as structure-from-motion (SfM) [31] have made it possible to use overlapping images captured with drones to create 3D reconstructions of the features present in the remotely sensed scene [28,30,31].

Common data products generated from point clouds are the digital surface model (DSM; elevations of features), digital terrain model (DTM; elevations of bare ground), canopy height model (CHM), and stitched orthomosaic [28]. The DSM and DTM are 2D images generated from the ground and non-ground classification of the point cloud [28,32]. The CHM is a 2D image representing the height of the features in the scene derived from

the DSM and DTM [28,32]. Two-dimensional orthomosaic images that consist of attributes such as reflectances are generated using the DSM and the individual images taken with the drone [28].

The point cloud and the 2D images containing the height information (i.e., DSM, DTM, and CHM) are often used for extracting the structural properties of remotely sensed objects such as the position and height [30]. In drone-based remote sensing studies of forests, the height information is commonly used for individual tree detection and crown delineation (herein referred to as "tree segmentation") [27,29,30]. In most studies of tree health to date, the 3D data are only used in an intermediary step (i.e., for tree segmentation) [29,30].

Given the extent of forest insect outbreaks and their probable increase in the future due to climate change, there is a need for advancing mapping methods that address not only tree mortality but also other types of damage such as top-kill and branch flagging (i.e., dead or discolored branches [19]). Such methods will aid tree health monitoring and attribute damage to different biotic agents, including bark beetles and defoliators. Despite the increasing use of drones in studies of insect outbreaks, very few studies have combined point cloud data with spectral or thermal data to assess the vertical profiles of trees. Most drone studies assessing tree health in forests have used the 2D multispectral orthomosaic to visually assess the study site and extract the spectral properties of the features (e.g., trees) present in the remotely sensed scene [27,30].

Some studies have used both spectral and structural information to produce 2D representations of tree heath [27]. Cardil et al. [33] used SfM-derived point cloud data for tree segmentation but used only spectral information to classify the pine processionary moth (*Thaumetopoea pityocampa*)-related defoliation levels of trees using vegetation indices with overall accuracies (OA) of 82–87%. Abdollahnejad and Panagiotidis [34] combined a texture analysis from a CHM (derived from a point cloud) and spectral information in a support vector machine model to classify the tree species and tree health status. Combining the texture analysis and spectral information improved the tree species classification but not the tree health status classification [34]. Cessna et al. [35] extracted structural metrics (such as the height and 3D grouping of points) from an SfM point cloud and spectral information from the multispectral orthomosaic in a study of spruce beetles. The trees were identified using on-screen manual digitization, and the authors developed models that mapped tree health [35]. The models that combined spectral and structural information or models that used only structural information had higher accuracies (OA: 75–77%) than models that used spectral information only (OA: 55–62%), highlighting the value of using structural information from point clouds in the assessment of tree health [35]. Lin et al. [36] used hyperspectral and thermal data and a point cloud to assess the capability of detecting the vertical profile of stress caused by pine shoot beetles in individual trees. However, no study has combined multispectral reflectances and a point cloud to produce the 3D health statuses of individual trees and characterized the resulting top-kill and tree damage across a landscape.

To address the above research need, we examined the utility of combining 3D and multispectral drone data to detect damage on parts of individual conifer trees in a forested area infested by insects in western Montana. We developed a novel approach to detect tree damage using multispectral reflectance values assigned to the 3D point cloud. We then calculated the amount of damage for each tree and analyzed the vertical location of the damage to identify the presence and estimate the length of top-kill. Our specific objectives were as follows:

1.  Segment individual trees within the study area using a multispectral SfM point cloud;
2.  Develop, evaluate, and apply a classification of points into healthy (green) or damaged (gray and red) classes based on their multispectral reflectances (point-level classification);
3.  Develop, evaluate, and apply an algorithm for identifying the percent damage, damage severity, and top-kill metrics of individual trees using the 3D classification of reflectances (tree-level damage algorithm).

## 2. Materials and Methods

### 2.1. Study Area

The study area was part of a broader project assessing tree damage using remote sensing. The site selection was based on an inspection of tree damage caused by defoliators and bark beetles from USFS ADS data and existing satellite imagery. An area that experienced outbreaks of defoliators and bark beetles over the last 15 years, based on aerial detection survey (ADS) data, was identified on Sheep Mountain, east of Missoula, Montana (46°57'34.3"N, 113°46'17.1"W; Figure 1 and Figure S1). The defoliators included the western spruce budworm (WSBW) and Douglas-fir tussock moth (DFTM), and the bark beetles included the mountain pine beetle (*Dendroctonus ponderosae*) and Douglas-fir beetle (*Dendroctonus pseudotsugae*). The study area has an elevation of approximately 1676 m, with an annual average temperature of 7 °C and annual average precipitation of 358.4 mm [37]. Common tree species in the study area are the subalpine fir (*Abies lasiocarpa*), lodgepole pine (*Pinus contorta*), and Douglas-fir (*Pseudotsuga menziesii*).

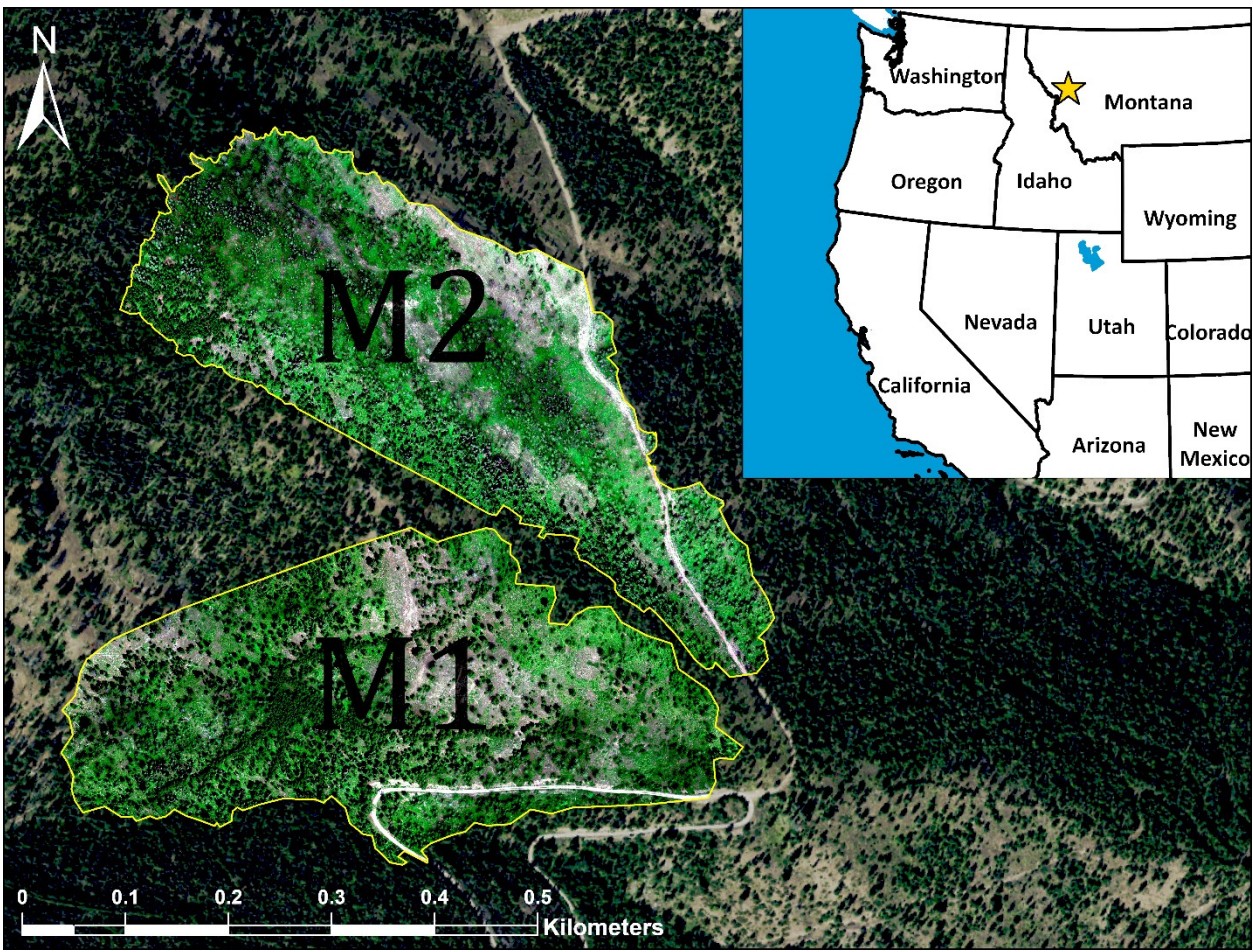

**Figure 1.** Study area near Missoula, Montana, USA. Yellow polygons define the clipped area used in this project (M1 and M2 are named after the two drone missions). The star on the inset map represents the study location in the northwestern US (basemap from ESRI). True color orthomosaic imagery within the yellow polygons is from the drone-mounted MicaSense MX-RedEdge sensor; outside of the yellow polygons, the basemap imagery is from the USDA National Aerial Inventory Program.

### 2.2. Drone Imagery Collection

Drone imagery was collected during two missions on 27 June 2022, at about 12 p.m. (M1) and 1:30 p.m. (M2) local time (approximately solar noon) over an area of 27 ha, with a flight altitude of 90 m and calm wind conditions throughout the missions (Figure 1). We

flew a DJI Matrice 210 V2 (dual gimbal) drone mounted with two sensors, a MicaSense MX-RedEdge sensor (AgEagle Aerial Systems, Inc., Wichita, KS, USA; herein referred to as "MS sensor") and a Zenmuse XT2 RGB sensor (SZ DJI Technology Co., Ltd, Shenzhen, China; herein referred to as "RGB sensor"), for image acquisition. The MS sensor unit is a radiometrically calibrated 5-band sensor capable of collecting data in the blue ($475 \pm 10$ nm; center wavelength and bandwidth), green ($560 \pm 10$ nm), red ($668 \pm 5$ nm), red edge ($717 \pm 5$ nm), and near-infrared ($840 \pm 20$ nm) wavelengths [38]. The sensor has five 1.2-megapixel imagers (one per band) with global shutters, which have a ground sampling distance (GSD) of 8 cm at a flight altitude of 120 m and a GSD of 4 cm at a flight altitude of 60 m [38]. The RGB sensor collects data in the visible wavelengths of the electromagnetic spectrum with a rolling shutter (12 M effective pixels) at a finer spatial resolution than the MS sensor and is not radiometrically calibrated.

The mission flights consisted of a single-grid flight path with an 80% frontal overlap and 75% side overlap for each image, with an average flight altitude of 91 m (Figure S1). The flight parameters were established with the objectives of simultaneous image acquisition with sufficient overlap from both sensors and maintaining a consistent altitude above the canopy, which approximately follows the topography. The elevation range within the mission areas was ~200 m. Fourteen ground control point (GCP) markers were placed across the drone mission areas to aid in the orthorectification and coregistration of data products between the RGB and MS sensors. The location of each GCP was recorded with a Trimble Geo X7 GPS receiver (Trimble, Inc., Westminster, CO, USA).

*2.3. Drone Imagery Pre-Processing*

The images collected from the RGB and MS sensors were used to generate two dense point clouds and two orthomosaics (one set for each sensor) in Agisoft Metashape Professional (v1.8.4 build 14671; herein referred to as "Metashape") using SfM photogrammetry. The point cloud and orthomosaic from the RGB sensor and the orthomosaic from the MS sensor were used to identify reference points and trees and also used for the qualitative inspection of the results (i.e., as guides). The point cloud from the MS sensor was used in the classification and subsequent analysis of the tree damage.

The SfM approach in Metashape creates an estimated 3D model that represents the orientations and locations of individual cameras in a 3D space [31,39]. An SfM reconstruction of the (initial) estimated 3D model of the study area is dependent on the detection and matching of the unique points (tie points) occurring in overlapping images [28]. This estimated 3D model is created using the image alignment workflow in Metashape, which allows the user to set the accuracy of the 3D model construction [39]. The accuracy parameter in Metashape determines if the images are upscaled, downscaled, or kept original for the identification and matching of unique points [39]. For this study, the "*high*" accuracy parameter was used for the SfM reconstruction of the initial estimated 3D model; this accuracy parameter retains the original size of the images [39]. Studies assessing the optimal parameter settings for tree detection and segmentation have suggested the use of the "*high*" setting [40,41].

The estimated 3D models from the RGB and MS sensors (one model for each sensor) were optimized with GCPs following the procedure recommended by Metashape [39]. Low-quality tie points with high reconstruction uncertainty and high reprojection errors were identified and removed using the "*gradual selection*" tool in Metashape, and the estimated 3D models were reoptimized [42]. The final dense point cloud and DEM were generated from the initial estimated 3D models using the "*high*" quality setting, similar to the accuracy parameter for image alignment (described above) [39]. Depth maps are intermediate products that estimate the 3D position of each point relative to individual images and are derived from the interior and exterior camera position parameters [39]. Multispectral values from the images from the MS sensor were assigned to each point in the final point cloud (herein referred to as "point cloud") with the "*Calculate Point Colors*" tool in Metashape using the depth maps [39]. Processing reports were generated from

Metashape before and after model optimization to report the improvement in positional errors (root mean square error, RMSE).

The DEM generated from the RGB sensor was used to produce an orthomosaic at an ~2 cm spatial resolution with individual images captured with the RGB sensor (herein referred to as "RGB orthomosaic"), and the DEM from the MS sensor was used to produce the "MS orthomosaic" from the radiometrically calibrated multispectral images captured with the MS sensor (~6 cm spatial resolution) (Figure 2).

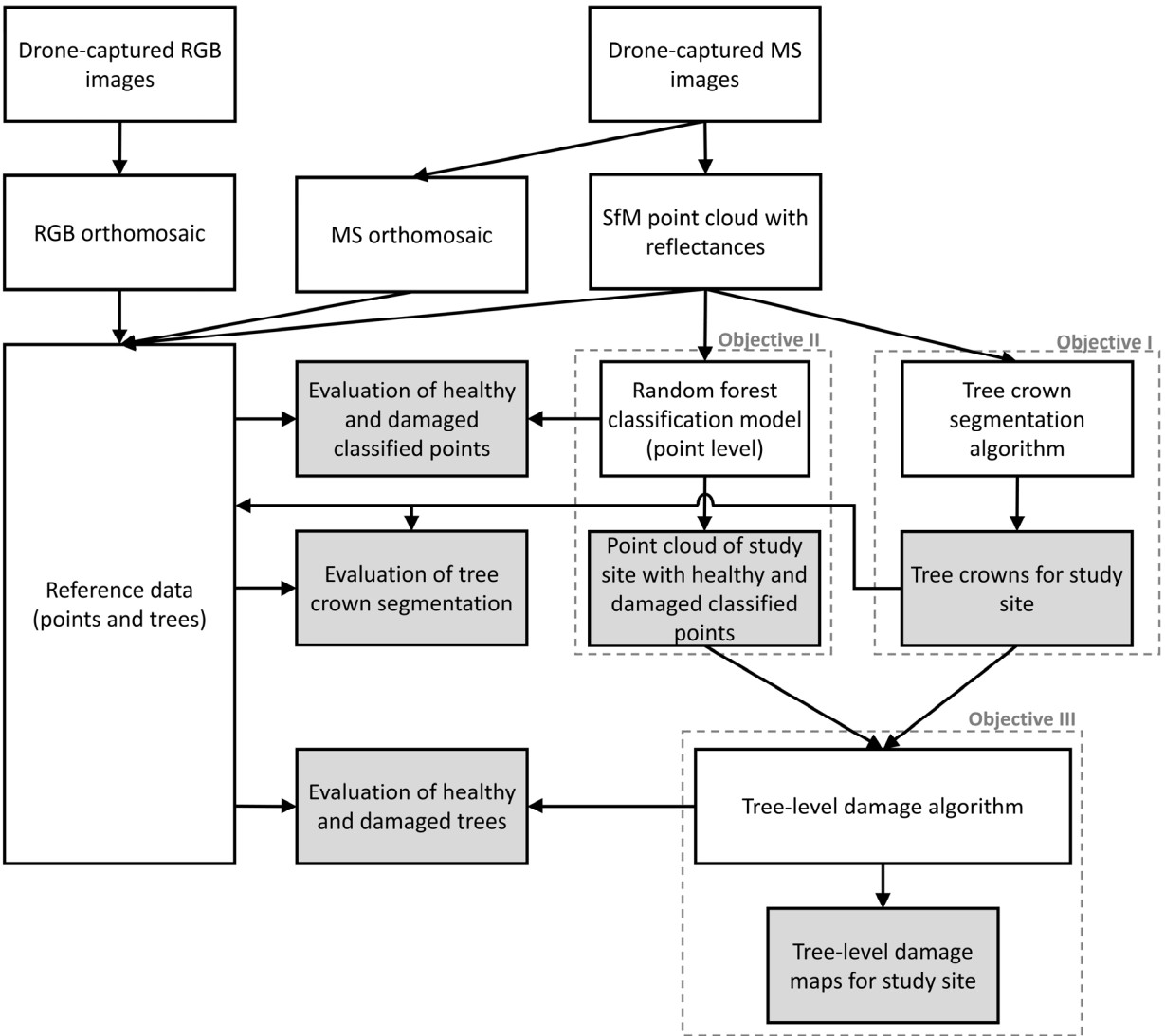

**Figure 2.** Project workflow for evaluating tree-level damage using structure-from-motion-derived (SfM) point cloud with multispectral reflectances and tree segmentation. Point-level classification includes healthy (green) and damaged (gray and red) classes. Boxes with dashed lines outline the objectives of this project. Boxes shaded gray represent evaluations and end products.

### 2.4. Reference Data

Following the approaches of multiple studies [33,35,43–45], an on-screen image inspection of the RGB (2-cm spatial resolution) and MS (6-cm spatial resolution) orthomosaics was used to assemble the reference data sets for training and evaluating the (a) tree segmentation, (b) the classification of individual points by health status, and (c) the separation of trees into healthy or damaged severities (Figure 2). We used the MS orthomosaic as the primary reference imagery and used the finer-resolution RGB orthomosaic as well as individual MS and RGB drone images to guide the selection of the reference data.

We generated a random set of 1000 locations across the study site for the on-screen identification of trees ("*tree*" class) and understory vegetation and ground ("*not tree*" class) and used this data set for evaluating tree segmentation.

For training the classification of individual points and the algorithm to label tree damage, we identified and manually labeled a set of 100 trees at random locations across the study area consisting of healthy trees, non-top-kill and top-kill trees with minor damage, moderate damage, major damage, and dead trees (Supplementary Materials Section S3.1) using an on-screen delineation of tree crowns with the MS and RGB orthomosaics and individual drone images. To evaluate the tree-level damage algorithm, we identified an additional random set of 1000 trees and recorded the damage severity of each tree; this independent set of trees was used for the evaluation.

For training and testing the classification of the point cloud reflectances, we identified 800 points that represented the health status of parts of trees (points in the point cloud), including healthy (green) and damaged (gray and red) classes (Supplementary Materials Section S3.2). These points were located in the reference data set of trees described above.

*2.5. Tree Segmentation*

2.5.1. Ground and Non-Ground Classification and Height Normalization of Point Cloud

We used a progressive morphological filter algorithm developed by Zhang et al. [46] to classify ground and non-ground points using the classify_ground function available in the lidR package [47]. The progressive morphological filter algorithm requires two parameters, a sequence of window size values (set to 0.75, 3, and 0.75 m) and a height threshold value (set to 2 m) (Figure S3a).

We used the normalize_height function from the lidR package [47] to generate a height-normalized point cloud from the points classified as ground or non-ground, thereby removing the influence of terrain [32] (Figure S3a). The "*algorithm*" parameter was set to the K-nearest neighbor with an inverse distance weighting algorithm using the knnidw function [47].

2.5.2. Point Cloud Segmentation into Unique Tree Objects

Image-based algorithms that use the CHM and point cloud-based algorithms that operate on individual points are common tools for segmenting individual trees in drone-based remote sensing of forests [30] (Supplementary Materials Section S4.2.1). We conducted a case study with a subset of data from the study area to compare an image-based algorithm by Silva et al. [48] with a point cloud-based algorithm by Li et al. [49] (Supplementary Materials Section S4.2.2, Figure S4). Based on a visual assessment, the point cloud-based segmentation algorithm by Li et al. [49] produced superior results for the case study. Therefore, the algorithm by Li et al. [49] was applied to the point cloud of the study area using the segment_trees function with the li2012 algorithm from the lidR package [47,49] (Figure S3b).

We used the crown_metrics function from the lidR package [47] with the point cloud, using tree segmentation as the input, resulting in a polygon feature data set of tree crowns. We performed an accuracy assessment of the tree segmentation using the reference data set of 1000 randomly generated locations identified as "*tree*" and "*not tree*" (described above), producing a confusion matrix, and computed the omission and commission errors of each class and the overall accuracy.

A second accuracy assessment of the tree segmentation was performed using the reference data set of 100 trees consisting of manually delineated tree crowns. This accuracy assessment compared areas of tree crowns from the tree segmentation algorithm and manually delineated tree crowns (reference data set) using Sørensen's coefficient (SC) [33,50]. *SC* is defined as:

$$SC = \frac{(2 * A)}{(2 * A) + B + C} \tag{1}$$

where $A$ is the area identified as the tree crown from both the manual delineation and the tree segmentation, $B$ is the tree crown area from the tree segmentation algorithm but

not the manual delineation, and *C* is the tree crown area from the manual delineation but not the tree segmentation algorithm (Figure S5) [33]. The *SC* value ranges from 0 to 1, where values close to 0 indicate a lower agreement and values close to 1 indicate a higher agreement [33,50].

### 2.6. Point-Level Classification with Random Forest Models

We developed models to classify the point cloud into green, gray, red, and shadow classes that represent the health statuses of the areas on trees that are healthy (green) and damaged (gray and red). We used the spectral reflectance values and indices of the reference data set of points (Supplementary Materials Section S3.2). Vegetation indices (Table 1) were calculated for each point of the reference data set.

**Table 1.** Name, equation, and reference for vegetation indices used in the project. ρ is reflectance.

| Name (Abbreviation) | Equation | Reference |
|---|---|---|
| Red–green index (RGI) | $\frac{\rho_{red}}{\rho_{green}}$ | Gamon and Surfus [51] |
| Simple ratio (SR) | $\rho_{NIR}/\rho_{red}$ | Woebbecke et al. [52] |
| Normalized difference vegetation index (NDVI) | $(\rho_{NIR} - \rho_{red})/(\rho_{NIR} + \rho_{red})$ | Rouse et al. [53] |
| Normalized difference red edge (NDRE) index | $(\rho_{NIR} - \rho_{rededge})/(\rho_{NIR} + \rho_{rededge})$ | Hunt Jr. et al. [54] |
| Green leaf index (GLI) | $\frac{(\rho_{green} - \rho_{red}) + (\rho_{green} - \rho_{blue})}{(2*\rho_{green}) + \rho_{red} + \rho_{blue}}$ | Hunt Jr. et al. [54] |
| Excess green (ExG) index | $(2*\rho_{green}) - \rho_{red} - \rho_{blue}$ | Woebbecke et al. [52] |
| Red–blue index (RBI) | $\rho_{red}/\rho_{blue}$ | Perez et al. [55] |
| Mean red–green–blue (meanRGB) index | $(\rho_{red} + \rho_{green} + \rho_{blue})/3$ | Clay et al. [56] |

Random forest (RF) classification is the most commonly used modeling tool in drone-based studies of forest insects [29,30]. RF models are nonparametric and nonlinear machine learning models with the ability to handle complex data with multiple predictor variables and higher dimensions [57,58].

One objective was to provide insights about important spectral regions that are useful for damage detection, thereby enabling the application of similar methods in different situations [59]. One relevant issue is multicollinearity, which can occur when evaluating explanatory variables [60]. High degrees of correlation are common in remote sensing data with spectral indices (Figure S7). From the set of 13 potential predictor variables (5 spectral bands and 8 spectral indices), a subset of predictor variables with low multicollinearity was selected (Figure S7). We used a "best subsets" approach for selecting the predictor variables in the final model. The best subsets approach is an iterative method that creates models from different combinations of potential predictor variables.

RF models were generated using the randomForest package, with 500 decision trees per model [61]. RF models classified each of the 800 points from the reference data set (described above) into green, gray, red, and shadow classes. The models were ranked based on their overall accuracy (OA), which is related to the out-of-bag error rate (OOB) (OA: 100% − OOB). To minimize the multicollinearity effects, models were only considered if the maximum pairwise absolute correlation coefficient between predictor variables was less than 0.7. Subsets of 1–4 predictor variables were considered. No five-variable subset had a maximum pairwise correlation of less than 0.7.

A set of the most accurate RF models was applied to a subset of the study site for further evaluation using the reference data set for trees. We visually compared the classified points on reference trees to the individual MS and RGB images. The RF model with a high OA and the best ability to capture the tree health condition was selected as the final RF model. We then applied this model to the point cloud of the entire study area.

*2.7. Tree-Level Damage Algorithm*

We developed an algorithm to separate trees into healthy versus different damage severities. This algorithm used the results of the tree segmentation and the point-level RF classification of healthy and damaged points as input (points classified as ground and shadows were excluded).

The algorithm performed a three-step procedure. First, it calculated the percentage of green, gray, red, and damaged points (sum of gray and red points) for each tree. Trees with less than 5% damage were identified as "*healthy*" trees, and the remaining trees were identified as "*damaged*" trees. We used the set of 1000 trees from the reference data (described above) to perform an accuracy assessment of the separation of healthy versus damaged trees. The algorithm was evaluated with a sampling (with replacement) of 200 trees per class (i.e., "*healthy*" and "*damage*" classes). The sampling was repeated 500 times, and the average values were reported in a confusion matrix.

In the second step, the "*damaged*" trees were separated into the following damage severities based on the percentage of damaged points: "*minor damage*" (5–25% damaged (red plus gray) points), "*moderate damage*" (25–75% damaged points), "*major damage*" (75–90% damaged points), and "*dead*" (>90% damaged points). The trees identified as "*dead*" were separated into "*dead (red)*" (>75% red points), "*dead (gray)*" (>75% gray points), and "*dead (mixed)*" (remainder of "*dead*" trees) (Figure S13). To evaluate the separation of healthy trees and those with different damage severities, the balance among the number of trees per class was improved by sampling (with replacement) the "*healthy*" (*n* = 75) and "*minor damage*" (*n* = 25) classes. The sampling was repeated 500 times, and the average values were reported in a confusion matrix.

In the third step, "*damaged*" trees were assessed for top-kill and identified as "*non-top-kill*" or "*top-kill*" using one of the two algorithms ("*bin2bin*" and "*top2bin*") described below. Top-kill trees are those that have continuous and nearly complete damage starting from the top of the tree extending downward [20]. For "*top-kill*" trees, we estimated the top-kill height (defined as the distance between the ground and the lowest top-kill bin), the length of top-kill (the distance in meters from the top of the tree to the bottom of top-kill), and the percentage of top-kill (top-kill length relative to the total tree height).

A visual inspection of the trees indicated variability in the crown characteristics of top-killed trees, and therefore, two top-kill algorithms were developed: the "*top2bin*" algorithm analyzed the cumulative damage from the treetop to the incremental bin, and the "*bin2bin*" algorithm analyzed the damage one bin at a time (Figure S14). The two top-kill algorithms use an iterative approach to evaluate damage at height intervals (bins) of 0.25 m, starting at the treetop and progressing downward by bin.

The "*top2bin*" algorithm considers the percentage of all damaged (red plus gray) points from the top of the tree to a given height bin, stopping when the percentage is <80%. The "*bin2bin*" algorithm assesses the percentage of damaged points within each height bin, stopping when the percentage within a bin is <90%. If the algorithms stop at the topmost bin, the tree is identified as "*non-top-kill*". The choice of implementation between the two algorithms is based on the percentage of damage points on a tree. An assessment of the reference trees with top-kill revealed that the stricter and more conservative "*bin2bin*" algorithm provided better top-kill estimates for trees with lower amounts of damage. Hence, the "*bin2bin*" algorithm was used for trees with less than 50% damaged points (of the total points for a tree), and the "*top2bin*" algorithm was used for the remaining trees.

The subset of trees identified as "*damaged*" from the reference data set of 1000 trees was used to evaluate the performance of the algorithm that identifies "*non-top-kill*" and

"*top-kill*" trees. The balance among the number of trees per class was improved by sampling (with replacement) the "*top-kill*" class (*n* = 40). The sampling was repeated 100 times, and the average values were reported in a confusion matrix.

*2.8. Characterization of the Extent of Tree Damage across the UAV Scene*

We applied the final RF model to the MS point cloud across the study area, resulting in green, gray, red, and shadow points. We also used the class probabilities computed by the RF model for each point based on the average proportion of votes for each class across all the trees in the random forest [57]. For example, if a point is classified as "*green*" in 439 out of 500 RF decision trees, the point will be classified as part of the "*green*" class, with a probability of (439/500 = 0.88). The estimated probabilities, which provide insights into classification confidences, are unrestricted by the assumptions about the data set (e.g., distribution) often required for regression-based models [62]. Class probabilities were averaged for all points within a tree.

We applied the tree-level damage algorithm using the tree segmentation results and the classified point cloud to compute the damage severity and metrics for each tree. The damage metrics included the percent green, percent gray, percent red, and percent damage (gray plus red) for each tree. The length of top-kill and percent top-kill were calculated for each top-kill tree.

To provide useful map products for scientific understanding and forest management, we spatially aggregated the tree-level damage metrics into 30 × 30-m resolution rasters. Centroids of tree segments (crowns) were calculated, and the mean damage metrics of trees (centroids) falling within each grid cell were computed.

**3. Results**

The drone imagery acquisition over the 27-ha study area yielded 8800 images from the MS sensor (one capture from the sensor stores five images, one per band; 1760 total captures) and 839 images from the RGB sensor. On a workstation with an Intel Xeon processor with 8 cores running at 3.5 Ghz and 32 Gb RAM, the pre-processing of the drone imagery took ~2.5 h to produce the point clouds for the study site. The processing time for the tree segmentation was 6 h, the application of the point classification model took 20 min, and the execution of the tree damage algorithm took 3 min. The root mean square error of the reprojection of the sparse point cloud (3D model) was 0.175 m. The 3D model optimization and GCP marker corrections reduced the average positional errors in X, Y, and Z from 1.85 m, 0.89 m, and 1.12 m to 0.02 m, 0.06 m, and 0.05 m, respectively, and reduced the total mean positional error from 2.43 m to 0.08 m (Table S1). The mean point density of the final 3D model was 62 points per square meter.

*3.1. Tree Segmentation*

The evaluation of the tree segmentation [49] resulted in an overall accuracy of 61.3% (Table 2). The commission error for the "*tree*" class was 12.4%, and the omission error was 58.7%. The commission error for the "*not tree*" class, which included cases when the algorithm detected a tree segment but no co-located reference tree was identified, was 49.1%, and the omission error was 8.8%. All the 352 cases of the reference trees misclassified as the "*not tree*" class were caused by the segmentation algorithm dividing a single tree into multiple trees (over-segmentation) combined with the reference location (identified as a tree) falling in between these over-segmented tree crown polygons (Figure S6). Of the 35 cases of the "*not tree*" reference locations misclassified as the "*tree*" class, 31 were due to the reference location being identified as the ground or the understory vegetation but falling within the erroneously delineated tree crown (Figure S6). Four cases were the misclassification of ground patches as trees. The second accuracy assessment comparing manually delineated and tree segmentation crown areas resulted in an SC of 0.79, which indicated a high agreement.

**Table 2.** Confusion matrix and accuracy metrics of the accuracy assessment of the point cloud-based tree segmentation algorithm. "*Tree segmentation issue*" refers to either the reference point (identified as a tree) falling in between crown segments of a single tree that was divided into multiple trees (over-segmentation) or the reference point (identified as on the ground or understory vegetation) falling within the erroneously delineated tree crown. "*Ground issue*" refers to the misclassification of understory or ground as trees.

| | Class | Reference | | Total | Commission Error (%) | User Accuracy (%) |
|---|---|---|---|---|---|---|
| | | Tree | Not Tree | | | |
| Prediction | Tree | 248 | 35 (31: tree segmentation issue; 4: ground issue) | 283 | 12.4 | 87.6 |
| | Not tree | 352 (352: tree segmentation issue) | 365 | 717 | 49.1 | 50.9 |
| | Total | 600 | 400 | 1000 | | |
| | Omission error (%) | 58.7 | 8.8 | | **Overall accuracy** | 61.3% |
| | Producer accuracy (%) | 41.3 | 91.2 | | | |

### 3.2. Point-Level Classification into Health Status Classes

The overall accuracy of the top classification models of the health statuses of points increased from one- to four-variable models (Figure S8). A substantial increase in the accuracy occurred from one- to two-variable models, a minor increase occurred from two- to three-variable models, and no increase occurred from three- to four-variable models (Figure S8). Therefore, only the two- and three-variable models were considered further.

The top five two-variable models had very similar results in terms of their overall accuracies (Table S2). Therefore, only the top-ranked two-variable model was used for comparison with the top three-variable models. The top-ranked two-variable model and the top five three-variable models were similar in terms of their overall accuracies (97.25–98.75%) (Table S2). Case studies of several reference trees were evaluated to further assess the differences among models (Figure S9). The classification of points by the second-ranked three-variable model (RBI + NDVI + REDEDGE model) best represented the damage locations of the reference trees (Figure S9). This model had an overall accuracy of 98.6%, an out-of-bag error of 1.4%, and class omission and commission errors of 0–3.5%, very similar to the highly ranked three-variable models (Table 3 and Table S2). Given the high accuracy and best performance in the visual inspection, the RBI + NDVI + REDEDGE model was selected as the final model and used in subsequent assessments (Figure S9). Points with reflectances that were more ambiguous in terms of health status had reduced classification probabilities reported by the random forest model (Figure 3), although the analysis of the case studies and selection of the final model minimized potential errors.

Indices computed using only RGB bands, such as RGI, RBI, meanRGB, and GLI were common among the top performing two-variable and three-variable models, and the most common spectral band was the green band (GRE) (Table S2). Models with predictor variables derived from only visible bands had a high accuracy (>97% for two-variable models and >98.5% for three-variable models). In addition, the inspection of reference trees (Figure S9) indicated that the RBI + GRE + GLI model captured damaged locations well.

**Table 3.** Confusion matrix and accuracy metrics of the final random forest model (RBI + NDVI + REDEDGE) for classifying points into healthy (green), damaged (red or gray), and shadow classes.

| | Class | Reference | | | | Total | Commission Error (%) | User Accuracy (%) |
|---|---|---|---|---|---|---|---|---|
| | | **Green** | **Gray** | **Red** | **Shadow** | | | |
| **Prediction** | Green | 199 | 1 | 2 | 2 | 204 | 2.0 | 98.0 |
| | Gray | 0 | 199 | 5 | 0 | 204 | 2.0 | 98.0 |
| | Red | 1 | 0 | 193 | 0 | 194 | 1.0 | 99.0 |
| | Shadow | 0 | 0 | 0 | 198 | 198 | 0 | 100 |
| | Total | 200 | 200 | 200 | 200 | 800 | | |
| | Omission error (%) | 0.5 | 0.5 | 3.5 | 1.0 | **Overall accuracy**: 98.6% | | |
| | Producer accuracy (%) | 99.5 | 99.5 | 96.5 | 99.0 | Out-of-bag error rate: 1.4% | | |

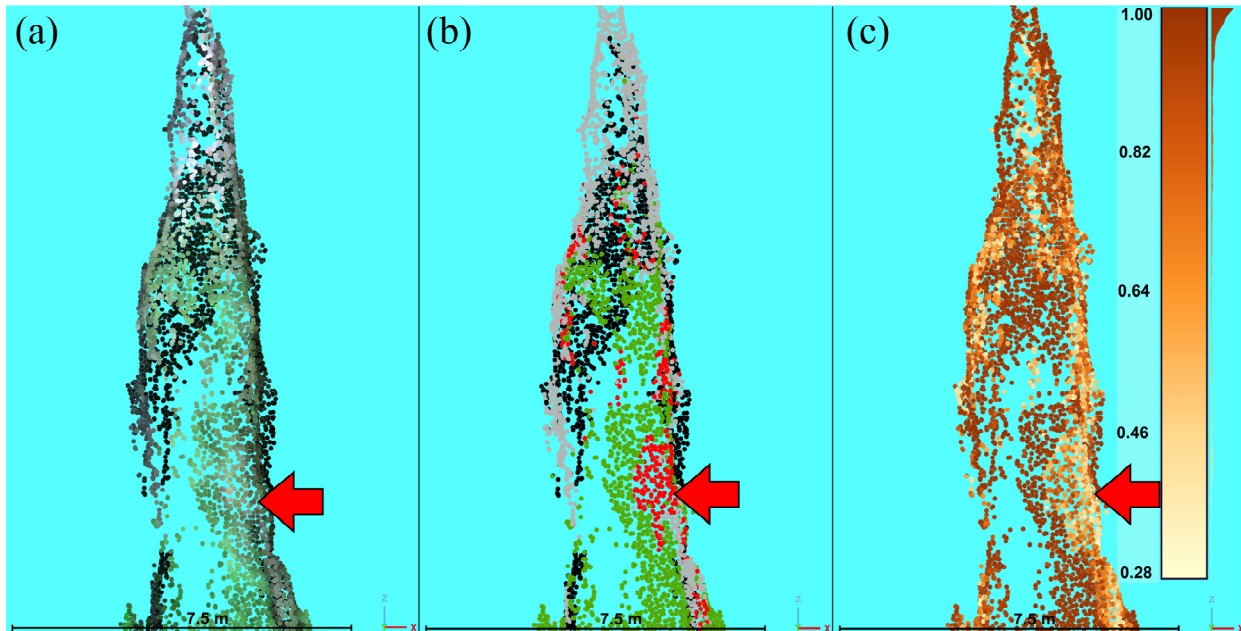

**Figure 3.** Classification probability (from application of the random forest (RF) model) of an example tree (profile view) that has parts of the crown with a more ambiguous health status (red arrow). (**a**) True color representation of the point cloud. (**b**) RF classification: green is healthy, red is red, gray is gray, and black is shadow. (**c**) The probabilities of classes shown in (**b**); darker colors represent higher probabilities of classification.

### 3.3. Tree-Level Damage

The results of the algorithm to separate "*healthy*" and "*damaged*" trees had an overall accuracy of 93.5% (Table 4). The omission and commission errors of the classes ranged from 2–11%. The results of the algorithm to separate healthy trees and different types of tree damage had an overall accuracy of 70.2% (Table 5), with class omission and commission errors ranging from 0–100%. The greatest confusion was between the "*dead*" classes ("*red*", "*gray*", and "*mixed*") and the "*major*" and "*moderate*" damage severities, which resulted in high omission and commission errors for all the "*dead*" sub-classes. High omission errors of 60–100% were likely caused by errors that occurred during the construction of the point cloud for trees lacking foliage (as discussed below in Section 4). The results of the algorithm to separate the trees identified as "*damaged*" trees into "*top-kill*" and "*non-top-kill*" had an overall accuracy of 91.8% (Table 6). The omission and commission errors of these classes

ranged from 5–14%. Figure 4 shows an example of branch flagging on a tree, in which only a portion of the tree's crown was gray (in this case, red parts of a crown might also be detectable).

**Table 4.** Confusion matrix and accuracy metrics for evaluation of algorithm to separate healthy trees from damaged trees. Results are averages from resampling (with replacement, 500 times) to address class imbalances.

| Tree Condition | Reference | | Total | Commission Error (%) | User Accuracy (%) |
|---|---|---|---|---|---|
| | **Healthy** | **Damaged** | | | |
| Healthy | 196 | 22 | 218 | 10.1 | 89.9 |
| Damaged | 4 | 178 | 182 | 2.2 | 97.8 |
| Total | 200 | 200 | 400 | | |
| Omission error (%) | 2.0 | 11.0 | | **Overall accuracy** | 93.5% |
| Producer accuracy (%) | 98.0 | 89.0 | | | |

*(Row label at far left, rotated: Prediction)*

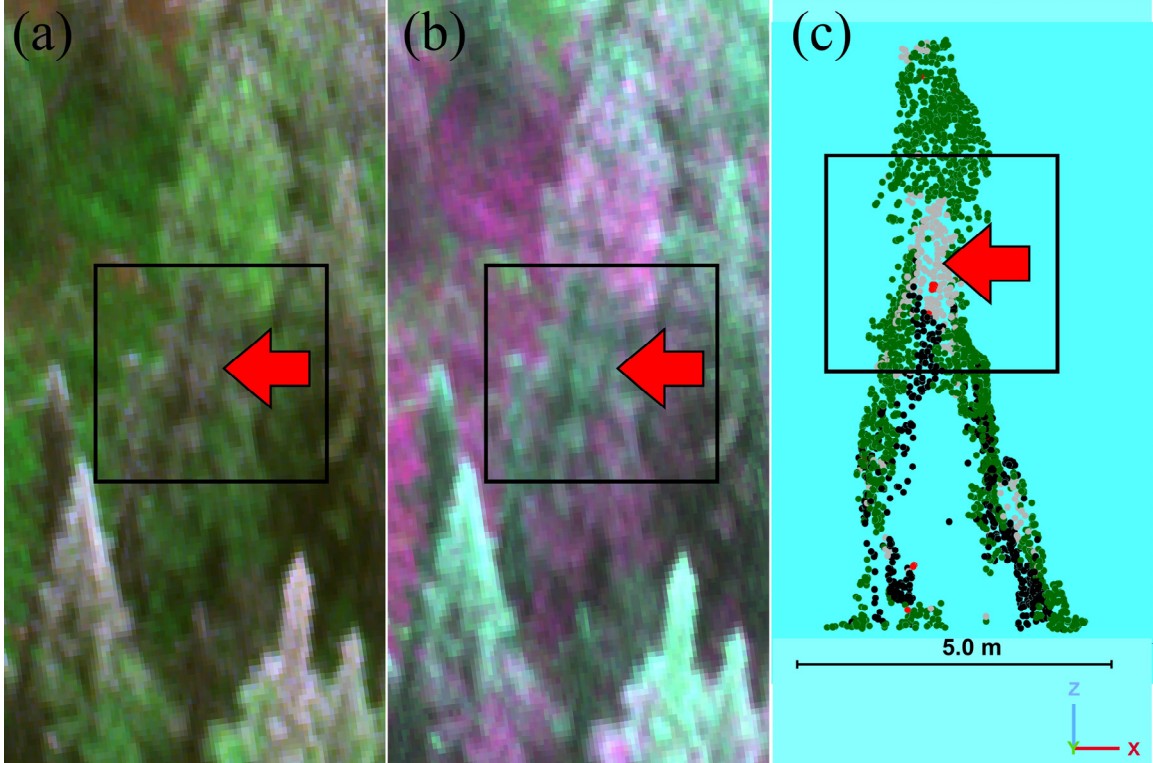

**Figure 4.** Example of branch flagging observed in a tree. Branch flagging is highlighted with rectangular box and red arrow. (**a**) True color image (individual drone image; map view). (**b**) False color image (individual drone image; map view). (**c**) Random forest model classification (profile view): green points are healthy, red is red, gray is gray, and black is shadow.

**Table 5.** Confusion matrix and accuracy metrics for evaluation of algorithm to separate healthy trees and different damage severities. Results are averages from resampling (with replacement, 500 times) to address class imbalances.

| | Damage Severity | Reference | | | | | | | Total | Comm. Err. (%) | User Acc. (%) |
|---|---|---|---|---|---|---|---|---|---|---|---|
| | | Healthy | Minor Damage | Moderate Damage | Major Damage | Dead (Red) | Dead (Gray) | Dead (Mixed) | | | |
| **Prediction** | Healthy | 74 | 4 | 6 | 0 | 0 | 0 | 0 | 84 | 11.9 | 88.1 |
| | Minor damage | 1 | 21 | 14 | 1 | 0 | 0 | 0 | 37 | 43.2 | 56.8 |
| | Moderate damage | 0 | 0 | 42 | 5 | 6 | 11 | 4 | 68 | 38.2 | 61.8 |
| | Major damage | 0 | 0 | 0 | 10 | 2 | 7 | 4 | 23 | 56.5 | 43.5 |
| | Dead (red) | 0 | 0 | 0 | 0 | 1 | 0 | 0 | 1 | 0.0 | 100.0 |
| | Dead (mixed) | 0 | 0 | 0 | 0 | 0 | 0 | 5 | 5 | 0.0 | 100.0 |
| | Total | 75 | 25 | 62 | 16 | 9 | 18 | 13 | 218 | | |
| | Omis. Err. (%) | 1.3 | 16.0 | 32.3 | 37.5 | 88.9 | 100.0 | 61.5 | | **Overall accuracy** | |
| | Prod. Acc. (%) | 98.7 | 84.0 | 67.7 | 62.5 | 11.1 | 0.0 | 38.5 | | 70.2% | |

**Table 6.** Confusion matrix and accuracy metrics for evaluation of top-kill algorithm to separate "*top-kill*" and "*non-top-kill*" trees within the subset of "*damaged*" trees. Results are averages from resampling (with replacement, 100 times) to address class imbalances.

| | Damage Type | Reference | | Total | Commission Error (%) | User Accuracy (%) |
|---|---|---|---|---|---|---|
| | | Non-Top-Kill | Top-Kill | | | |
| **Prediction** | Non-top-kill | 18 | 2 | 20 | 10.0 | 90.0 |
| | Top-kill | 3 | 38 | 41 | 7.3 | 92.7 |
| | Total | 21 | 40 | 61 | | |
| | Omission error (%) | 14.3 | 5.0 | | **Overall accuracy** | 91.8% |
| | Producer accuracy (%) | 85.7 | 95.0 | | | |

### 3.4. Tree Damage across the Study Site

The segmentation process identified a total of 15,519 trees across the study site (Table 7). Most of the trees were classified as "*healthy*" (78.3%) and the remaining were "*damaged*" (21.7%). A majority (58.7%) of the damaged trees showed "*minor damage*", that is, trees with 5–25% in damage points (red plus gray). Among "*damaged*" trees, most were identified as "*top-kill*" (78.9%); top-kill trees had an average top-kill length of 1.5 m (17.8% of the total tree height on average) (Table 5). The "*dead (mixed)*" class had the highest average length of top-kill (5.8 m), and the "*dead (gray)*" class had the highest average percentage of top-kill (66.3% of the total tree height).

The RF model applied to the point cloud of the study site reported the classification probability for each point, with 77% of points having a classification probability of 0.75 or higher, indicating that most points were classified with a moderate to high classification probability. Most points in the green or gray classes were classified with a high classification probability (Figure S12a). Points in the red or gray classes exhibited a range of probabilities greater than 0.4 (Figure S12b,c).

**Table 7.** Summary of tree-level metrics of healthy and damaged trees, with different damage severities across the study site. "TK": top-kill.

| Tree Type | No. of Trees | Mean % Green | Mean % Gray | Mean % Red | Mean % Damage | No. of Non-TK | No. of TK | Mean TK Length (m) | Mean % TK |
|---|---|---|---|---|---|---|---|---|---|
| Healthy | 12,143 | 99.4 | 0.4 | 0.3 | 0.7 | - | - | - | - |
| Damaged | 3376 | 72.1 | 8.3 | 19.6 | 27.9 | 713 | 2663 | 1.5 | 17.8 |
| Minor damage | 1980 | 87.6 | 4.1 | 8.4 | 12.4 | 541 | 1439 | 0.9 | 11.1 |
| Moderate damage | 1192 | 56.3 | 11.9 | 31.9 | 43.8 | 169 | 1023 | 1.7 | 19.7 |
| Major damage | 154 | 18.3 | 27.3 | 54.4 | 81.7 | 3 | 151 | 4.7 | 54.7 |
| Dead (red) | 5 | 5.3 | 91.2 | 3.5 | 94.7 | 0 | 5 | 4.8 | 39.8 |
| Dead (gray) | 19 | 4.6 | 7.5 | 87.9 | 95.4 | 0 | 19 | 4.0 | 66.3 |
| Dead (mixed) | 26 | 5.0 | 41.9 | 53.1 | 95.0 | 0 | 26 | 5.8 | 57.9 |
| Total (all trees) | 15519 | 93.4 | 4.5 | 2.1 | 6.6 | - | - | - | - |

The results of the tree segmentation and classification showed a good visual agreement when overlaid on the MS orthomosaic (Figure 5). The trees identified as "*healthy*" had green foliage, and the trees identified with having different damage severities displayed less green and more gray or red in the crowns (Figure 5a). The trees identified as "*top-kill*" with top-kill percentages less than 75% (relative to height of the tree) were apparent with gray tops and green in the lower half of the crown for trees, and the trees identified as "*top-kill*" with greater than 75% top-kill were predominantly gray, red, or mixed in the MS orthomosaic (red arrows in Figure 5b). The probabilities of classifying the points averaged for each tree were higher on healthy trees (0.78–0.98) compared to damaged trees (0.58–0.68) (Figure 5c).

The maps of the study area illustrate the spatial patterns of damage for tree-level (Figure 6a,b,d) and 30 m resolution (Figure 6c,e) data. For this study area, trees exhibiting moderate to high damage and higher top-kill percentages (51–100%) occurred in distinct clusters (Figure 6). Most trees were classified with a moderate to high probability (mean tree-level probabilities of 0.58–0.98) (Figure 6b). The southern mission (M1) had higher classification probabilities than the northern mission (M2), because M1 was comprised of more healthy trees compared with M2, and healthy trees, constituting of predominantly "*green*" classified points, have a higher classification probability than other classes (Figure S12a, Figures 5c and 6b).

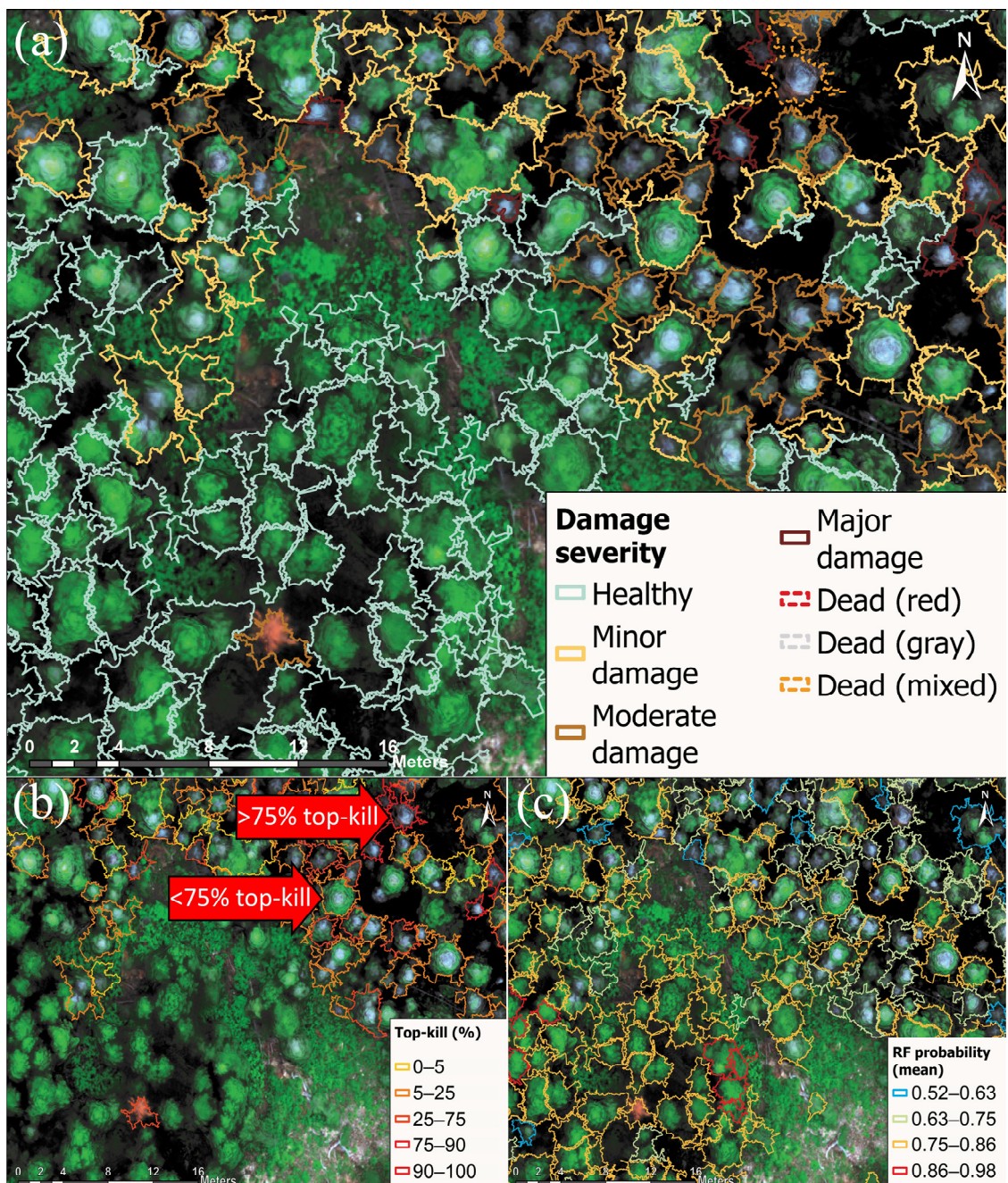

**Figure 5.** Zoomed-in views of tree-level damage metrics for part of the study site. The basemap imagery is the true color drone orthomosaic (collected using a MicaSense RedEdge MX sensor). Polygons delineate the crowns of individual trees. (**a**) Damage severity. (**b**) Percentage of top-kill length relative to tree height for trees identified as "*top-kill*". Trees identified as "*healthy*" or "*non-top-kill*" are not outlined. Labeled red arrows show examples of damaged trees with different degrees of top-kill. (**c**) Average random forest ("RF") probability of point classifications for each tree. Lower bounds of ranges for legends in (**b**,**c**) are exclusive, and upper bounds are inclusive, except for the first range, where both bounds are inclusive.

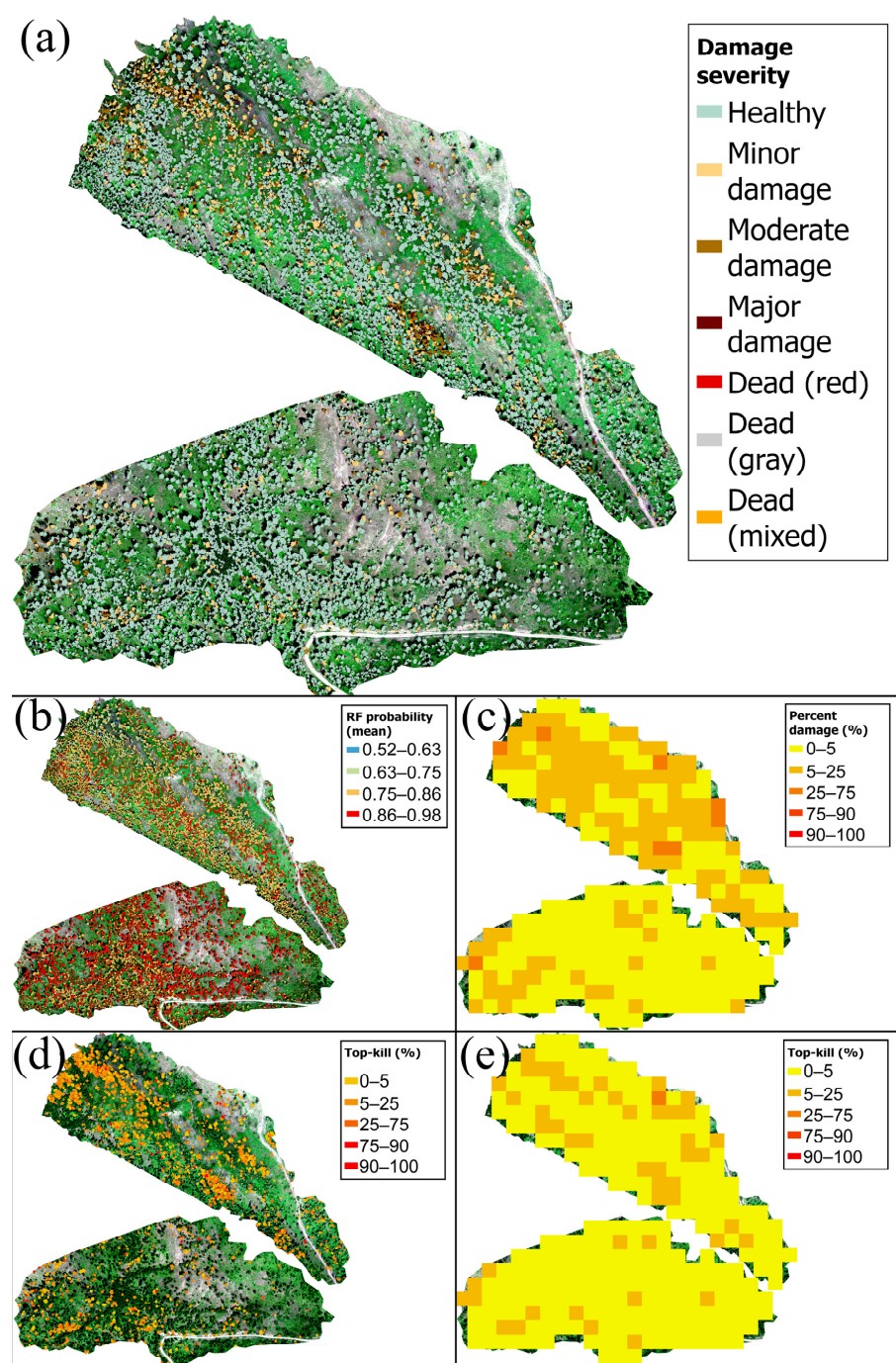

**Figure 6.** Tree-level damage metrics for the study area displayed as tree-level maps (**a**,**b**,**d**; polygons cover the crown of each tree) and spatially aggregated maps (**c**,**e**; 30 m spatial resolution). Basemap imagery is the true color drone orthomosaic (collected using a MicaSense MX-RedEdge sensor). (**a**) Damage severity of each tree. (**b**) Average probability of random forest ("RF") classification for all points in each tree. (**c**) Mean percentage of damage (red plus gray) for trees within the grid cell. (**d**) Percentage of top-kill relative to tree height for trees identified as "*top-kill*". Trees identified as "*healthy*" or "*non-top-kill*" are not outlined. (**e**) Mean percentage of top-kill relative to the tree height within the grid cell. Lower bounds of ranges for legends in (**b**–**e**) are exclusive, and upper bounds are inclusive, except for the first range, where both bounds are inclusive.

## 4. Discussion

We developed a novel approach that analyzed the tree health status in a 3D environment by classifying each point in a multispectral point cloud, thereby considering the vertical location and extent of damage within a tree crown. We then computed tree-level damage metrics, including top-kill. Although past studies incorporated structural information when mapping insect-caused damage [34,35], they did not assess the vertical distribution of damage, which allows for a more refined estimate of tree damage.

The performance of the tree segmentation in this study was similar to other studies that used point clouds [63–65]. The accuracies of tree-level damage algorithms are dependent on the performance of the tree segmentation. The accuracy assessment indicated that the primary issue was not in identifying trees; instead, most of the issues arose from the algorithm dividing trees into multiple segments (over-segmentation); only 4 out of 1000 accuracy assessment reference locations were misclassified as ground. Additionally, the use of Sørensen's coefficient (SC) showed high spatial agreement between the areas of tree crowns from the reference data set and the crowns delineated by the tree segmentation.

The final RF model that classified points was chosen based on the model's overall accuracy (OA) and a visual assessment of the ability to capture the health conditions within the crowns of example trees. The final RF model included RBI, NDVI, and REDEDGE as predictor variables and had a high OA. The NDVI and REDEDGE spectral indices are commonly used in vegetation detection, especially in tree health studies [12,15,30]. Some sensors mounted on drones and satellites do not have red edge or near-infrared bands. Models using predictor variables (indices and bands) from only visible bands also yielded high accuracies, indicating the effectiveness of this methodology without needing NIR, REDEDGE, and related indices and the potential application using drones with only RGB sensors.

The RF model classified points into "*green*", "*gray*", and "*red*" classes, with classification probabilities ranging from moderate to high. A visual assessment of the reference trees indicated that points with lower classification probabilities occurred in localized areas on the tree crown, with a mix of green, gray, and red points (an example shown in Figure 3). Difficulties in assessing damage within highly localized areas on tree crowns occur in field surveys as well. Studies of observer errors from field surveys assessing defoliation and discoloration [66] and overall tree health [67] have reported inconsistencies in needle damage estimates due to factors such as the number of surveyors present [66] and view angle during the assessment [67]. In fact, a drone image analysis can complement existing field and aerial survey data to address some of the sources of inconsistency among observations. Collecting images from above with a consistent ~nadir view angle at low altitudes enables the drone-based sensor to capture details of top–down damage in individual tree crowns that cannot be viewed from the ground and are too fine grained to be detected using satellite or airborne sensors [68]. Multiple observers can subsequently review the same set of images to quantify variability or bias among observers [69]. Automated classification and the ability to archive or re-analyze imagery (as opposed to single field or aerial observations) make the monitoring workflow more reproducible [69] as well as scalable to provide a more even coverage of larger areas, with potential to quantify change over time [68,69].

The first step of the tree-level damage algorithm, which separated "*healthy*" and "*damaged*" trees, had a high overall accuracy; the second step, which separated "*damaged*" trees into different damage severities, had a reduced accuracy (as expected). Remote sensing studies mapping healthy and damaged trees using fine-resolution drone imagery have reported similar high overall accuracies [43,45,70]. Furthermore, similar moderate overall accuracies have been reported when mapping different damage severities, including the classification of the tree-level health status [71,72] and estimation of the damage severity [33,73].

Some confusion occurred between the "*minor damage*" and "*moderate damage*" classes. The misclassification of "*minor damage*" trees as "*moderate damage*" trees (and vice versa) was

due to the trees being slightly under or over the percent damage threshold that separates the two damage severities, together with some uncertainty in visually defining the reference damage severities for these trees. We tested the model with different percent damage thresholds and found no overall improvement.

Based on an inspection of the classified points for trees identified as "*dead*" in the reference data set, cases of confusion between the "*dead*" trees ("*red*", "*gray*", and "*mixed*") and "*moderate*" and "*major*" damaged trees were likely caused by the presence of green points in the middle or bottom half of the "*dead*" trees (Figure S15). The SfM algorithm searches for reoccurring features in multiple drone images to estimate a feature's (e.g., a branch's) position in the three-dimensional space [31,39]. In the case of "*dead*" trees (which can be missing substantial amounts of foliage), the algorithm could have experienced difficulties in separating the relatively thin branches of the tree of interest from the herbaceous understory or foliage of surrounding trees. Multispectral values are assigned to points using depth maps estimated from the captured images with the "*Calculate Point Colors*" tool in Metashape [39]. The multispectral values are not directly measured; hence, there could be difficulties in accurately assigning multispectral reflectances to areas such as the thin branches of snags. As a result, the reflectance values from the herbaceous understory or foliage of surrounding trees might have been misassigned to points in the "*dead*" trees, leading to green points being present in the "*dead*" trees. Similar difficulties in classifying "*dead*" trees have been reported in studies using computer vision-based approaches to classify tree damage in two-dimensional products [74,75].

The third step of the tree damage algorithm separated all "*damaged*" trees into "*non-top-kill*" and "*top-kill*" with a high accuracy. As an outcome of this algorithm, we considered calculating the uncompacted live crown ratio, a metric used for the field-based assessment of the tree crown health [76], computed as the length of live crown relative to the height of the tree. However, estimating crown base heights in this study proved challenging. Qualitative visual assessments of SfM point clouds of trees did not reveal distinct features that could be used to identify the base of the live crown. The absence of points near the crown base in the SfM point cloud did not consistently correspond to the crown base when compared with drone images (Figure S16). This absence of points could be attributed to the occlusion of the crown base by neighboring trees in multiple drone images, resulting in an insufficient number of tie points (recurring features in multiple images), and therefore, failure in constructing points for the feature (crown base) [31,39]. The issue of identifying the crown base height could be addressed by using lidar point clouds [77,78], leveraging lidar's capability to penetrate through the canopy and discern understory details, an aspect that might be hindered in SfM point cloud construction of crown base and understory structures due to the occlusion of the respective features in individual images.

The application of the tree-level damage algorithm across the study area identified most trees as "*healthy*" and the rest as "*damaged*". Most of the "*damaged*" trees in the study area were "*top-kill*" trees with an average damage of less than a quarter of the tree from the treetop downward. "*Dead*" trees had the largest average length of top-kill, spanning more than three-quarters of the height of the tree from the treetop downward. Although not a focus of this study, the methods developed here might be useful in identifying branch flagging on individual trees, as some trees that showed signs of branch flagging on the drone-captured images had similar patterns of damage on the respective point clouds of the health status-classified points (Figure 4). Tree-level damage maps of the study area showed clusters of tree damage, demonstrating the feasibility of utilizing a point cloud to map tree damage and produce tree-level damage metrics useful for monitoring (Figures 5 and 6).

The spatially aggregated maps produced in this study demonstrate the capability to generate spatially upscaled data products, moving from a tree-level scale to a medium-resolution satellite data scale (e.g., Landsat). This spatial upscaling facilitates the integration of tree-level products from drones with satellite data, which is useful for the mapping of forests [79] and development and testing of ecological models to inform management decisions [80]. The results produced in this study demonstrate the ability of drones to

bridge the gap between field-based forest inventory methods and space-based remote sensing of forests to aid in research and management: a role that has been highlighted in multiple reviews of forest remote sensing [23,79,81].

## 5. Conclusions

We developed a novel, accurate method for detecting insect-caused damage on parts of individual trees from a point cloud and the reflectances acquired using a drone. We used this 3D classification to map trees as healthy and having different damage severities across the study site and estimate the top-kill characteristics. Top-kill is characteristic of certain forest insects, such as the DFTM and WSBW, and might be used to differentiate damage from defoliators from bark beetles [12,19,22–24]. Thus, these methods can not only refine estimates of where the damage is located on individual trees but also have the potential for improving the attribution of damage to insect species or types.

The algorithms developed here from drone imagery can be applied using point clouds derived from data from other platforms as well. Recent studies have utilized NAIP imagery in constructing point clouds for forestry inventory estimates [82,83]. In addition, the methodology can be applied using multispectral lidar systems that simultaneously collect active remote sensing data in multiple wavelengths [84,85]. However, it may be important to consider the financial implications of acquiring lidar sensors and lidar-capable systems. Exploring the capability of SfM point clouds using less expensive sensors was one of the motivations behind this project.

Additional future directions could focus on ecological applications in management and research. Results of applying the methods of mapping three-dimensional tree damage described here can be used to examine impacts of snow interception [86,87], habitat availability [88,89], tree canopy volume estimations [90], and carbon stocks [91].

The data products and maps produced in this study demonstrate how a structure-from-motion-derived point cloud from drone imagery can be used to map 3D damage within trees and estimate top-kill. The methods developed here advance our understanding of the potential advantages and limitations of using drones to monitor and map forest insect and disease damage.

**Supplementary Materials:** The following supporting information can be downloaded at: https://www.mdpi.com/article/10.3390/rs16081365/s1, The Supplementary Materials include information on the study area and drone data pre-processing, reference data assembly, tree segmentation algorithms, point classification models, and tree damage assessment and top-kill detection algorithms, which are described in 2. Materials and Methods section and referred to in 3. Results section. The Supplementary Materials also contain examples of cases of errors and limitations of the methodology presented in the study, which are described in 4. Discussion section. References [92–97] are also cited in Supplementary Materials file. The following titles of figures and tables have been summarized from the titles present in the Supplementary Materials document: Figure S1. Drone flights superimposed on polygons from the USDA Forest Service ("USFS") Region 1 ("R1") Aerial Detection Surveys ("ADS") showing insect damage from recent years; Figure S2. Methods for assembling the reference data set of point clouds using manual point picking in CloudCompare; Figure S3. Height normalization and tree segmentation results for a subset of the point cloud; Figure S4. Comparison of segmentation results from raster-based and point cloud-based algorithms; Figure S5. Examples of polygons used in ArcGIS Pro to compute Sørensen's coefficient (SC) for the evaluation of tree segmentation; Figure S6. Examples of cases of errors during random placement of evaluation reference points for the evaluation of tree segmentation; Figure S7. Correlation matrix of predictor variables used in random forest models; Figure S8. Overall accuracy (OA) as a function of the number of variables used in the random forest (RF) models; Figure S9. Examples of trees used to evaluate classification models of healthy versus damaged points; Figure S10. Pairwise plots of predictor variables used to develop the final random forest model (RBI + NDVI + REDEDGE) using reference data set of points; Figure S11. Final random forest model applied to one of the drone sites; Figure S12. Distributions of classification probabilities for different classes of tree crown and shadow health representations; Figure S13. Classification tree used for the separation of trees into healthy, damaged, and different

damage severities; Figure S14. Diagrammatic representation of the top-kill assessment algorithms used in the study; Figure S15. Example of issues with damage-classified point cloud of dead trees, with points classified as "green" present in the middle and bottom; Figure S16. Example of limitation of photogrammetric point clouds in determining live crown base; Table S1. Estimated positional errors (root mean square error, RMSE) from each processing step by drone mission; Table S2. Top five two-variable and three-variable random forest models.

**Author Contributions:** Conceptualization, A.S., J.A.H., A.J.H.M., J.W.K. and A.T.S.; Methodology, A.S., J.A.H., A.J.H.M., J.W.K. and A.T.S.; Investigation, A.S., J.A.H., A.J.H.M., J.W.K. and A.T.S.; Writing—original draft, A.S., J.A.H., A.J.H.M., J.W.K. and A.T.S.; Supervision, A.S. and J.A.H.; Funding acquisition, J.A.H. and A.J.H.M. All authors have read and agreed to the published version of the manuscript.

**Funding:** This work was supported by a NASA SmallSat Program Award (80NSSC21K1155) and the US Department of Agriculture, National Institute of Food and Agriculture, and the McIntire Stennis project 1019284.

**Data Availability Statement:** Code and sample data (subset data from the study area M2) for this project are available through the following GitHub repository: https://github.com/abhinavshrestha-41/Drone_PointCloudClassification_TopkillDetection (accessed on 15 February 2024).

**Acknowledgments:** We greatly appreciate the work performed by our field crew: Luke Schefke (Washington State University), Andrew Hudak (US Forest Service), and Benjamin Bright (US Forest Service).

**Conflicts of Interest:** The authors declare no conflicts of interest.

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
