# Peer review of "Evaluating a Novel Approach to Detect the Vertical Structure of Insect Damage in Trees Using Multispectral and Three-Dimensional Data from Drone Imagery in the Northern Rocky Mountains, USA"

_remotesensing, doi:10.3390/rs16081365_

Round 1

Reviewer 1 Report

Comments and Suggestions for Authors

What is the accuracy and resolution of the 3D model created using MetaShape?

In the flowchart you have three boxes without outputs.  If those are end state conditions, they should be at the bottom of the chart.

Equation 1. You define C in the text, but have a C prime in the equation.  Are these the same?

L426 Formatting error.

Table 2 requires reformatting.  Difficult to see which entry goes on which line.

Reviewer 2 Report

Comments and Suggestions for Authors

The paper appears to be based on an MSc thesis using drone-based aerial photogrammetry and multispectral imaging to assess tree health from data collected in Montana, USA. The author(s) used the multispectral reflectance values combined with the 3D point cloud data processed from images that were acquired simultaneously from a RGB camera that was mounted on a dual gimbal onboard a DJI M210 drone.

The paper is well-written, with no issues with regards to use of the English language. The abstract and introduction are strong, with sufficient information in the materials & methods sections to describe the project and data collection.

Is there any discussion on wind speed and changes in direction over the course of the field surveys? Were there trees moving in the wind while the drone was collecting data? The pixel density in Figure 4 makes it difficult to see detail. Is a ground sampling distance of 8 cm enough spatial resolution to get sufficient detail of the trees to evaluate them?

How does the assessment of the tree health using the drone and algorithm classification system compare to an assessment of tree health and insect damage from the ground? Is the remote sensing method proving accurate?

Table #3 seems to have an issue with the formatting of “Predict” on the left side of the table.

The paragraph starting on Page 17 (line 564) is excessively long and difficult to follow. Can this be restructured using bullet points for Figure descriptions or is this a document error?

The discussion about using lidar as an assessment tool in the future is interesting and would be a great follow-up study if the equipment can be utilized.

The paper is thorough, with much explanation of the algorithms used and their purposes.

A few additional artifacts were noticed in document and need to be edited:

P12L427: “).” at the start of the paragraph

Table #4: fix “Prediction” word wrapping in first column

P17L540: “).” at the start of the paragraph

P17L564: suggest restructuring paragraph

Reviewer 3 Report

Comments and Suggestions for Authors

Title, Abstract, Keywords:

 OK.

Introduction:

The introduction is well written, and the objectives are well defined. The references used are relevant. 

Materials and Methods, Results:

Both the methods and the results are very well described. Although the article is quite extensive, it does not contain unnecessary parts. A large scope is devoted to describing methods and results, but their detail is useful for the reader, especially for cases of performing similar analyses.

 Discussion and Conclusions:

 OK.

Supplementary Materials:

They are quite extensive, but help to understand the method and clearly show intermediate results. I also rate them very positively.

References:

 OK.

The authors developed a new method to detect insect-caused damage on parts of individual trees from a point cloud and reflectances acquired by a drone. They combined multispectral reflectances and a point cloud to produce the 3D health status of individual trees and detected the top-kill and tree damage. The results of this study can be useful in practice and other similar analyses. In my opinion, the manuscript is written very well and I have only the following minor comments:

1. References to figures and tables are missing in the text (automatic references probably failed).

2. In lines 565-607, the explanations for Figure 6 are repeated several times (probably also a technical problem).

3. In Figure 1, it is convenient to use the km designation instead of "Kilometers".

In Figure 4, it would be better to use rounded values in the scale (eg 5 m).

Figures 5 and 6: In the legend, the end and start numbers of the two following intervals should not be the same (e.g. instead of 0-5; 5-25 it could be 0-5; 5.1-25, etc.).

But these are rather cartographic details. Other maps and figures are good.

I congratulate the authors on this article and wish it to be published soon!
